Effects of Origanum vulgare essential oil and its two main components, carvacrol and thymol, on the plant pathogen Botrytis cinerea

http://orcid.org/0000-0001-8787-7785 Hou Huiyu 1 2 3
Zhang Xueying 1 2 3
Zhao Te 1 2 3 tezhao@126.com
Zhou Lin 1 2 3 zhoulinhenau@163.com
1 Henan Key Laboratory for Creation and Application of New Pesticides , Zhengzhou, Henan , China
2 Henan Research Center of Green Pesticide Engineering and Technology , Zhengzhou, Henan , China
3 College of Plant Protection, Henan Agricultural University , Zhengzhou, Henan , China
Wink Michael
Electronic publication date: 2020 Aug 14
Publication date: 2020
Volume: 8
Electronic Location ID: e9626
Received 2020 Feb 5; Accepted 2020 Jul 8
Copyright: © 2020 Hou et al.
Copyright year: 2020
Copyright holder: Hou et al.
License: This is an open access article distributed under the terms of the Creative Commons Attribution License, which permits unrestricted use, distribution, reproduction and adaptation in any medium and for any purpose provided that it is properly attributed. For attribution, the original author(s), title, publication source (PeerJ) and either DOI or URL of the article must be cited.
License URL: https://creativecommons.org/licenses/by/4.0/

Keywords: Origanum vulgare, Carvacrol, Thymol, Botrytis cinerea, Antifungal activity, Botanical fungicide

Funding: Innovation Scientists and Technicians Troop Construction Projects of Henan Province 134100510009 National Natural Science Foundation of China 31371962 The study was financially supported by the Innovation Scientists and Technicians Troop Construction Projects of Henan Province (134100510009) and the National Natural Science Foundation of China (31371962). The funders had no role in study design, data collection and analysis, decision to publish, or preparation of the manuscript.

==============================
Background

Botrytis cinerea causes serious gray mold disease in many plants. This pathogen has developed resistance to many fungicides. Thus, it has become necessary to look for new safe yet effective compounds against B. cinerea.

Methods

Essential oils (EOs) from 17 plant species were assayed against B. cinerea, of which Origanum vulgare essential oil (OVEO) showed strong antifungal activity, and accordingly its main components were detected by GC/MS. Further study was conducted on the effects of OVEO, carvacrol and thymol in vitro on mycelium growth and spore germination, mycelium morphology, leakages of cytoplasmic contents, mitochondrial injury and accumulation of reactive oxygen species (ROS) of B. cinerea. The control efficacies of OVEO, carvacrol and thymol on tomato gray mold were evaluated in vivo.

Results

Of all the 17 plant EOs tested, Cinnamomum cassia, Litsea cubeba var. formosana and O. vulgare EOs had the best inhibitory effect on B. cinerea, with 0.5 mg/mL completely inhibiting the mycelium growth of B. cinerea. Twenty-one different compounds of OVEO were identified by gas chromatography–mass spectrometry, and the main chemical components were carvacrol (89.98%), β-caryophyllene (3.34%), thymol (2.39%), α-humulene (1.38%) and 1-methyl-2-propan-2-ylbenzene isopropyl benzene (1.36%). In vitro experiment showed EC50 values of OVEO, carvacrol and thymol were 140.04, 9.09 and 21.32 μg/mL, respectively. Carvacrol and thymol completely inhibited the spore germination of B. cinerea at the concentration of 300 μg/mL while the inhibition rate of OVEO was 80.03%. EC50 of carvacrol and thymol have significantly (P < 0.05) reduced the fresh and dry weight of mycelia. The collapse and damage on B. cinerea mycelia treated with 40 μg/mL of carvacrol and thymol was examined by scanning electron microscope (SEM). Through extracellular conductivity test and fluorescence microscope observation, it was found that carvacrol and thymol led to increase the permeability of target cells, the destruction of mitochondrial membrane and ROS accumulation. In vivo conditions, 1000 μg/mL carvacrol had the best protective and therapeutic effects on tomato gray mold (77.98% and 28.04%, respectively), and the protective effect was significantly higher than that of 400 μg/mL pyrimethanil (43.15%). While the therapeutic and protective effects of 1,000 μg/mL OVEO and thymol were comparable to chemical control.

Conclusions

OVEO showed moderate antifungal activity, whereas its main components carvacrol and thymol have great application potential as natural fungicides or lead compounds for commercial fungicides in preventing and controlling plant diseases caused by B. cinerea.

Introduction

Botrytis cinerea, a necrotrophic plant pathogenic fungus with broad hosts, can cause gray mold disease in many plants. The infected plants include many important agricultural, economic and horticultural crops, with a total of more than 1,400 species such as grape, strawberry, tomato, cucumber, orchid and other fruits, vegetables and flowers (Fillinger & Elad, 2016). It not only infects field crops, but also causes huge losses to the crops after harvesting (Romanazzi & Feliziani, 2014). At present, benzimidazoles, dicarboximides, carbamates and antibiotics are the main fungicides to control this disease, whereas the resistance of B. cinerea to these fungicides has been widely reported (Leroux et al., 2002; Bardas et al., 2010; Liu et al., 2019). Most field investigations have demonstrated that B. cinerea has developed resistance to carbendazim, procymidone and diethofencarb in tomato and blueberry, and the point mutations, such as carbendazim-resistant phenotypes carried the point mutation E198A in the β-tubulin gene (tub) and procymidone-resistant phenotypes carried the point mutations I365S and a pair of point mutations Q369P and N373S in the amino acid sequence (Bos1), have been identified (Adnan et al., 2018; Sautua et al., 2019). Therefore, the control of gray mold is still facing great challenges. It is of great significance to explore new antifungal components for the integrated management of gray mold.

Origanum species are herbaceous perennials distributed in different parts of the world including the Mediterranean, Central Asia, the Arabian Peninsula, Northern Africa, and Europe (Aligiannis et al., 2001; De Martino et al., 2009a). The genus Origanum consisits of 43 species and 18 hybrids (Tepe, Cakir & Sihoglu Tepe, 2016). They are typically applied in food and cosmetics as a flavoring and aromatic agent (Gomez et al., 2018; Jan et al., 2020), and widely used in agriculture and traditional medicine due to its antimicrobial activity and antioxidant activities (Rodriguez-Garcia et al., 2015; Lu et al., 2018; Elshafie et al., 2015). Because of different extraction techniques, plant parts or collected regions, different constituents were found in Origanum species, but most of them contain phenols, terpenes and their derivatives, such as γ-terpinene, α-thujene, β-myrcene, α-terpinene, terpinen-4-ol, β-caryophyllene, linalool, limonene, thymol and carvacrol (Mamadalieva et al., 2017; Khan et al., 2019; Karpinski, 2020; Ebadollahi, Ziaee & Palla, 2020).

Previous studies have shown that the EOs of Origanum species (such as O. compactum, O. heracleoticum, O. majorana, O. onites, O. vulgare subsp. hirtum and O. syriacum var. bevanii) exhibited antimicrobial activities against a variety of plant pathogenic fungi and bacteria (Table 1). In addition, they had excellent effects on the control of agricultural insect pests (Gong & Ren, 2020; Pavela, 2012), stored product insects (Ebadollahi, Ziaee & Palla, 2020) and acarids (Koc et al., 2013; Zandi-Sohani & Ramezani, 2015; Shang et al., 2016). Interestingly, although many of Origanum species shared the same ingredients, it showed significant differences to the same microorganism (De Martino, De Feo & Nazzaro, 2009b; Mamadalieva et al., 2017; Moumni et al., 2020). This suggests that the antimicrobial effect of EOs may be related to the proportion of the main antimicrobial components or the synergistic with antagonistic effects of different chemical components (Wink, 2015; Tepe, Cakir & Sihoglu Tepe, 2016). So far, there is no evidence that Origanum species and its main components are harmful to humans and animals and have high phytotoxicity (Mancini et al., 2014; Llana-Ruiz-Cabello et al., 2017; Elshafie et al., 2017).

Table 1 A list of reports indicating antimicrobial effects of EOs isolated from the Origanum genus.

Origanum species	Chemotype	Antifungal	Antibacterial	Ref.	
O. vulgare	Carvacrol, thymol	Botrytis cinerea,
Fusarium solani var. coeruleum	Clavibacter michiganensis subsp. Michiganensis,
Propionibacterium acnes
Staphylococcus epidermidis	(Daferera, Ziogas & Polissiou, 2003; Taleb et al., 2018)	
O.vulgare. Subsp. hirtum	Carvacrol, thymol	Monilinia laxa,
Monilinia fructigena,
Monilinia fructicola	Escherichia coli,
Listonella anguillarum,
Vibrio alginolyticus,
Vibrio splendidus,
Vibrio sp.,
Saccharomyces cerevisiae	(Stefanakis et al., 2013; Elshafie et al., 2015)	
O. compactum	Carvacrol, thymol	Alternaria alternata,
Bipolaris oryzae,
Fusarium graminearum,
Fusarium equiseti,
Fusarium verticillioides		(Santamarina et al., 2015)	
O. heracleoticum	Carvacrol	Aspergillus niger, Botrytis cinerea, Monilinia fructicola,
Penicillium expansum	Bacillus megaterium,
Clavibacter michiganensis,
Xanthomonas campestris,
Pseudomonas fluorescens,
Pseudomonas syringae pv. phaseolicola	(Della Pepa et al., 2019)	
O. majorana	terpinen-4-ol,
δ-2-carene,
υ-Terpinene,
carvacrol	Aspergillus niger,
Monilinia fructicola,
Penicillium expansum	Bacillus megaterium,
Bacillus subtilis,
Clavibacter michiganensis,
Escherichia coli,
Listonella anguillarum,
Pseudomonas aeruginosa,
Pseudomonas fluorescens,
Pseudomonas syringae pv. phaseolicola,
Saccharomyces cerevisiae,
Salmonella enterica,
Staphylococcus aureus,
Vibrio splendidus,
Vibrio alginolyticus,
Vibrio sp.,
Xanthomonas campestris	(Della Pepa et al., 2019; Moumni et al., 2020; Stefanakis et al., 2013)	
O. onites	Carvacrol, thymol	Botrytis cinerea,
Colletotrichum acutatum,
Colletotrichum fragariae,
Colletotrichumg loeosporioides,
Phomopsis obscurans	Bacillus subtilis,
Escherichia coli,
Listonella anguillarum,
Vibrio splendidus,
Vibrio alginolyticus,
Vibrio sp.,
Saccharomyces cerevisiae	(Altintas et al., 2013; Stefanakis et al., 2013)	

Although extensive studies of the EOs of Origanum species have been conducted, however, O. vulgare populations from the Asian area have been poorly explored. In addition, many kinds of EOs, such as Cinnamomum cassia (Ma et al., 2019), Litsea cubeba (Yang et al., 2010) and Zanthoxylum leprieurii (Tanoh et al., 2020), etc., have also great potential in the control of plant disease. Moreover, the researches on Origanum EOs and its main components against plant pathogens mainly focus on the antimicrobial activities (De Martino et al., 2009a; Stefanakis et al., 2013), while the mechanism of antifungal action is still unclear and the actual application is also necessary to be evaluated. Thus, this study was conducted to determine: (i) the antifungal activity of 17 plant EOs against B. cinerea, (ii) the main components of Origanum vulgare essential oil (OVEO) by GC-MS, (iii) the antifungal activities and mechanism of OVEO, carvacrol and thymol on B. cinerea in vitro, (iiii) in vivo control effects of OVEO on B. cinerea, providing a foundation for the development and its utilization as a plant fungicide.

Materials and Methods

Inhibitory activity of different plant EOs on mycelium growth of B. cinerea

Seventeen plant EOs were involved in the current study, namely O. vulgare, C. cassia, Perilla frutescens, Saussurea costus, Mentha spicata, L. cubeba, Asarum sieboldii, Illicium verum, Foeniculum vulgare, Angelica dahurica, Curcuma zedoaria, Mentha haplocalyx, Artemisia argyi, Eucalyptus globulus, Syzygium aromaticum, Acorus tatarinowii and turpentine (mixture) EOs. All the EOs were purchased from Cedar pharmaceutical Co., LTD., Jiangxi, China. Isolates of B. cinerea were prepared from tomato fruits (Luoyang, China) and deposited at the Key Laboratory of Creation and Application of Novel Pesticides, Henan. The preliminary antifungal activities of these EOs against B. cinerea were determined according to the method described by Farzaneh et al. (2015). Emulsion of the EOs were prepared by sterile water with 0.5% (v/v) acetone and Tween 80, blended with Potato Dextrose Agar (PDA, 200 g extract of boiled potatoes, 20 g dextrose and 20 g agar powder in 1,000 mL distilled water) at 40–45 °C to obtain final concentrations of 0 (control), 0.5 and 2 mg/mL. A five mm mycelial disc from young cultures of target fungi (3 d) was placed in treated sterile plates (D = 7 cm) after medium solidification. The petri dishes were sealed with sealing film and cultured upside-down in dark at 26.5 °C.

The mycelial growth was measured by a nonius using decussation way at 2, 4 and 6 d, respectively. Three replicates per treatment were used in each experiment and the experiments were performed three times. Growth inhibition of each EO was computed by the formula (1): (1) Inhibition(%)=(Dc−Dt)/(Dc−0.5)×100

Where Dc is the colony diameter of the control group; and Dt is the colony diameter of the treatment groups with EO.

GC–MS analysis

The chemical compositions of OVEO was analyzed by GC–MS (Agilent-7890B, Agilent Technology Co., LTD., USA) according to the method reported by Scalas et al. (2018). Chemicals in OVEO were separated with HP-5 MS capillary column (50 m × 0.25 mm × 0.25 μm), high-purity Helium (99.999%) as the carrier gas. The helium carrier gas rate was 1.0 mL/min at a split ratio of 20.4:1, and the injection volume was 1.0 μL. The initial temperature was set at 60 °C for 25 min, then gradually increased to 150 °C at a rate of 1.2 °C·min−1. Electron ionization was used as the ion source, and the ionization energy was set at 70 eV. The quadrupole temperature was 150 °C, and the ion source temperature was 230 °C. The sector mass analyzer was set to a range from 33 to 900 amu. Diluted samples (1/100, in acetone) at 1 μL each were injected and mass spectra were compared with the standard mass spectra from the NIST 2.2 database provided by the software of GC–MS system.

In vitro antifungal activities of OVEO, carvacrol and thymol against B. cinerea

OVEO, carvacrol (purity 99%) and thymol (purity 98.5%, Sigma Aldrich, St. Louis, MO, USA) were dissolved in sterile water containing 0.1% (V/V) acetone and Tween 80 to prepare the stock solution at the concentration of 1,000 μg/mL (the stock solution shall be used for dilution below unless otherwise specified). The fungi were grown on a PDA plate for 5 d, and then harvested using a five mm sterilized puncher along the edges of the colonies, and the fungal blocks were placed in the center of PDA plates containing 0, 2.5, 5, 10, 15, 20, 40, 50 μg/mL carvacrol and 0, 2.5, 5, 10, 20, 40, 60, 80 μg/mL thymol. Cultures with 0.005% (0.008%) acetone and Tween 80 were used as the control, respectively. Incubation upside-down at 26.5 °C. The fungistatic effect was observed, and the colony diameter was measured and recorded when the diameter of the control was exceeded 7.5 cm. Then, calculations were made for B. cinerea mycelium inhibition ability of OVEO, carvacrol and thymol and regression equation and EC50 values by probability value analysis. Three replicates per treatment were carried out in each experiment and the experiments were performed three times.

Antifungal activity of OVEO, carvacrol and thymol on spore germination of B. cinerea

B. cinerea isolates were cultured on PDA plates at 20 °C for 14 d. Then, the plates were injected with five ml sterile water and gently scraped with a sterile swab and filtered through four layers of sterile cheesecloth to obtain spore suspension (Jing et al., 2018). The spore suspension concentration was adjusted to 105~106 spores mL−1 by using a hemocytometer and continuously adding sterile water. OVEO, carvacrol and thymol were blended with acetone and Tween 80 and sterile water, respectively, and the concentration was adjusted to 100, 200, 300, 400, 500, 600 μg/mL at the same time, a 0.5 mL aliquot of conidium suspension of B. cinerea plus 0.5 mL liquid were added to a tube. Then, 60 μL was transferred onto a double concave glass plate with pipette gun and placed in a wet sterile culture dish. Cultures with 0.03% acetone and Tween 80 were used as the control. The cultures were incubated at 28 °C. After incubating for 8 h, spores of B. cinerea were observed with a microscope (Nikon Eclipse Ti-S, Tokyo, Japan) at a 200× magnification. The spore germination percentage was calculated by counting germinated spores among 200 spores. Three replicates per treatment were conducted in each experiment and the experiments were performed three times.

Effects of carvacrol and thymol on fresh and dry mycelial weight of B. cinerea

The fresh and dry mycelial weight of the tested pathogen treated with carvacrol and thymol were measured (Chen et al., 2014). According to the EC50 and EC90 of carvacrol and thymol, two components were dissolved and diluted in sterile water with 0.1% acetone and Tween 80, and transferred into Erlenmeyer flasks with 25 mL of Potato Dextrose Broth (PDB, 200 g extract of boiled potatoes, 20 g dextrose in 1,000 mL distilled water) medium to obtain carvacrol concentrations of 0 (control), EC50 (9.38 μg/mL) and EC90 (43.49 μg/mL) and thymol concentrations of 0 (control), EC50 (21.32 μg/mL) and EC90 (65.13 μg/mL), respectively. The pathogen was harvested along the edge of the colony (3 d) with a 5 mm sterilized punch, then inoculated into each flaskand the flasks were incubated at 120 r/min shaking and 26.5 °C. The mycelium groups were harvested on 3 d and strained off excess culture. The agar blocks were removed with tweezers, and each mycelium groups were washed with sterile water at least three times, and then filtered for 30 min to remove excess water on the surface. The fresh mycelia were weighed and the dry mycelia as well after drying at 60 °C for 12 h. Each treatment consisted of three replicates and the experiments were performed three times.

Effect of carvacrol and thymol on the relative leakage of B. cinerea

The permeability of the membrane was expressed in terms of relative extracellular conductivity and tested according to the method of Zhou et al. (2018) by using a Seven Excellence Multiparameter Tester (METTLER TOLEDO Instrument Co., Ltd., Shanghai, China) with some modifications. Carvacrol and thymol stock solution were prepared with sterile water and 10% dimethyl sulfoxide, and the final concentrations were 10, 200, 500 μg/mL and 70, 200, 500 μg/mL, respectively. Vinclozolin (500 μg/mL) was prepared with sterile water and 10% dimethyl sulfoxide as solvent. Fungal samples were collected 3 d after inoculation on PDA medium. Then the collected mycelium was inoculated in PDB and shaken at 120 rpm and 26.5 °C for 108 h. Remove the agar blocks with tweezers, wash each mycelium groups with sterile water at least three times, and filter for 30 min to remove excess water on the surface. The mycelia (0.1 g) were separately put into the solution with the above series concentrations into glass tubes of 10 mL, and 10% dimethyl sulfoxide (DMSO) sterile distilled water was the solvent control. Vinclozolin containing 10% DMSO was used as a positive control. Extracellular conductivity was measured at 0, 10, 40, 140, 220, 360 and 400 min. Finally, the electrical conductivity was measured again after dead treatment (temperature ≥ 95 °C, 25 min). Three replicates per treatment were carried out in each experiment and the experiments were performed three times. The relative leakage was determined according to the formula (2): (2) Relativeleakage(%)=(Ct−C0)/Cd×100

Where Ct is the conductivity of current time samples, C0 is the conductivity of initial time (0 min) samples, and Cd is the conductivity of dead treatment conductivity samples.

Scanning electron microscopy and fluorescent microscopy

Carvacrol and thymol were dissolved and diluted in sterile water with acetone and Tween 80. One mL of mixture and nine mL PDA medium were mixed and poured into a 9 mm diameter Petri dishes. The final concentrations of 0 (control), 25 and 40 μg/mL of carvacrol and thymol were obtained, respectively. Medium containing 0.004% acetone and tween-80 was used as solvent control. The blocks of B. cinerea were inoculated on the medicated medium and cultured upside down at 26.5 °C for 5 d. The effects of carvacrol and thymol on mycelial morphology were analyzed using the modified method of Li et al. (2017). For SEM observation, PDA blocks about 5 × 7 mm were cut from the edge of mycelium. Mycelium exposed to different treatments was fixed with 25% glutaraldehyde for 48 h at low temperature. Samples were washed for 3 h (10 min/time) in 0.1 molL−1 phosphate buffer (pH 7.2). After being fixed with 1% osmium for 2 h, the samples were washed with distilled water for three times (3 min/time) and dehydrated in a graded series of ethanol concentrations (30%, 50%, 70%, 80%, 90% and twice at 100%) for 15 min at each stage. Samples were dried by CO2 critical point drier (CPD 030, Leica, Wetzlar, Germany), then gold coated using a sputter coating machine (MC1000, Hitachi, Tokyo, Japan). All samples were viewed in a SEM (Model SU8010-3400N, Hitachi) operating at 10 kV at 2.5 k× magnification.

The fluorescence intensity of H2DCFDA dye under specific excitation wave is related to the content of ROS, and the accumulation of ROS can be judged by the fluorescence intensity (Li et al., 2017). SYTOX Green is a fluorescent dye commonly used to study membrane integrity. It can pass through damaged membrane cells but cannot penetrate the complete cell membrane and emit Green fluorescence. Rhodamine-123 is a cationic dye that can penetrate the cell membrane and is an indicator of mitochondrial transmembrane potential. Mitochondrial matrix fluorescence intensity weakened or disappeared in normal cells, and strong yellow-green fluorescence was released when mitochondrial membrane was destroyed (Tian et al., 2012).

Mycelium was observed by Nikon Eclipse Ti-S inverted fluorescence microscope (Eclipse Ti-S, Tokyo, Japan) and slightly modified (Jing et al., 2018). The fungal blocks were inoculated into 50 mL erlenmeyer flasks with PDB culture medium and the flasks were incubated at 120 r/min and 26.5 °C for 2 d. Then the samples were incubated in PDB with the carvacrol and thymol (100 and 200 μg/mL) for 12 h, 0.1% acetone and Tween 80, where sterile water was used as the blank control. The collected mycelia were stained with 1 μg/mL H2DCFDA, SYTOX Green or Rhodamine 123 for 30 min at 4 °C in darkness. After the dying, the mycelia were washed with phosphate buffered saline to remove the residual dye and then observed by fluorescence microscope. The views were randomly selected from each group, and all experiments were repeated six times and the experiments were performed three times.

Protective and therapeutic effects of OVEO, carvacrol and thymol on tomato gray mold caused by B. cinerea

Protective effect experiment: tomato fruits with same size and weight were selected and put in sterile plastic boxes (9 tomatoes in each box) with gauze at the bottom wetted with sterile water to keep moisture. Tomatoes were sprayed with OVEO, carvacrol or thymol at 500 and 1,000 μg/mL, respectively. Pyrimethanil at 400 μg/mL was used as the standard control, and 0.1% acetone plus Tween 80 with sterile water was used as the solvent control. Sterile water was used as the blank control. Each box had one treatment, nine repetitions for each treatment, and three fungal blocks for each repetition. After the liquid was dried, fungal blocks were placed on each tomato by acupuncture method and placed in the artificial climate box with humidity over 85% and temperature at 26.5 °C. After 48 h, the diameter of the spot was measured and the control effect was calculated. Therapeutic effect experiment: Similar to protection, the above mentioned tomato fruits were inoculated with fungal blocks and cultured under the above conditions. After 24 h, the above mentioned concentration solution was taken out and sprayed on the tomato fruits for further culture. After 48 h, the size of diseased spots was measured. The protective and therapeutic effects were calculated by the following formula (3): (3) Controlefficacy(%)=(Dc−Dt)/Dc×100

Where Dc is the diameter of disease spots in the control group; and Dt is the spot diameter in the treatment group.

Statistical analysis

The statistical software SPSS (V 20.0; Chicago, IL, USA) was used for all data analyses. Data were analyzed by Duncan’s multiple range test at the level P < 0.05.

Results

Inhibitory activities of 17 plant EOs on mycelial growth of B. cinerea

The antifungal activities of 17 plant EOs against B. cinerea are shown in Fig. 1. C. cassia, L. cubeba and O. vulgare EOs completely inhibited the mycelial growth at the concentration of 0.5 mg/mL. In addition, the inhibition rate of M. spicata and S. aromaticum EOs reached 91.70 and 87.55%, respectively. At the concentration of 2 mg/mL, the inhibition rate of M. haplocalyx, I. verum and A. sieboldii EOs was 100%, 99.56% and 82.53%, respectively.

Figure 1 Inhibitory activity of 17 plant EOs on the mycelial growth of B. cinerea.

(Columns for each EOs from left to right denote 0.5 and 2.0 g/L). The fungal mycelial growth diameter of the control and treatments with different concentrations EOs were measured after 6 d of cultivation. The data were expressed as mean ± SD of three replicates. Values of different letters in the same column were significantly different at P < 0.05. The X-axis refers to different EOs, and the Y-axis refers to the inhibition rate of mycelial growth.

GC–MS analysis of OVEO

In total, 21 different chemical components are identified from the OVEO (Table 2; Fig. 2). Carvacrol (89.98%) was found to be the major component, followed by β-caryophyllene (3.34%), thymol (2.39%), α-humulene (1.38%) and 1-methyl-2-propan-2-ylbenzene isopropyl benzene (1.36%). In addition, the isomers carvacrol and thymol had adjacent peaks.

Table 2 Chemical components of the OVEO determined by GC–MS.

No.	Retention time
(min)	Compound	Molecular
formula	Area (%)a	
1.	3.574	Butyl acetate	C6H12O2	0.04	
2.	4.106	Diacetone alcohol	C6H12O2	0.16	
3.	5.777	2-butoxyethanol	C6H14O2	0.03	
4.	6.532	α-thujene	C10H16	0.01	
5.	6.789	α-pinene	C10H16	0.07	
6	7.384	Camphene	C10H16	0.02	
7	8.649	β-pinene	C10H16	0.21	
8	9.404	β-myrcene	C10H16	0.07	
9	10.835	α-terpinene	C10H16	0.04	
10	11.338	1-methyl-2-propan-2-ylbenzene	C10H14	1.36	
11	11.590	(4R)-limonene	C10H16	0.34	
12	11.756	1,8-cineole	C10H18O	0.12	
13	13.038	(E)-β-ocimene	C10H16	0.02	
14	13.713	γ-terpinene	C10H16	0.10	
15	38.381	Thymol	C10H14O	2.39	
16	39.903	Carvacrol	C10H14O	89.98	
17	46.283	α-copaene	C15H24	0.08	
18	50.803	(-)-β-caryophyllene	C15H24	3.34	
19	54.071	(1E,4E,8E)-α-humulene	C15H24	1.38	
20	61.521	β-cadinene	C15H24	0.18	
21	66.596	(-)-Caryophyllene oxide	C15H24O	0.07	
Note:

a Relative proportions of EO constituents.

Figure 2 GC-FID chromatogram of OVEO by an HP-5MS column.

The characterized peaks are numbered according to the serial numbers in Table 2. The chemical formulae of carvacrol and thymol are indicated next to their peaks.

Antifungal activities of the OVEO, carvacrol and thymol against mycelial growth of B. cinerea in vitro

The effects of the OVEO and its main components on B. cinerea are shown in Table 3. Among the four main constituents, OVEO, carvacrol and thymol could significantly inhibit the mycelial growth of B. cinerea, where the effect increased with increasing the concentration, whereas β-caryophyllene showed no inhibitory effects on B. cinerea. The EC50 of carvacrol and thymol were 9.09 and 21.32 μg/mL, respectively, lower than that of OVEO (140.04 μg/mL). Carvacrol and thymol as main antifungal components have been demonstrated, and were chosen for further study.

Table 3 Toxicities of OVEO, carvacrol and thymol against mycelial growth of B. cinerea.

EO	Regression equation	r	EC50 (95% FL) (μg/mL)	χ2	
OVEO	Y = 1.7615 X + 1.2193	0.9740	140.04 (58.07~337.72)	7.55	
Carvacrol	Y = 1.9704 X + 3.1117	0.9931	9.09 (3.96~20.85)	3.72	
Thymol	Y = 2.6426 X + 1.4884	0.9974	21.32 (9.95~45.68)	3.99	
β-caryophyllene	–	–	–	–	

Antifungal activities of OVEO, carvacrol and thymol on spore germination of B. cinerea

Carvacrol and thymol at all tested concentrations (50–300 μg/mL) result a significant low germination of B. cinerea spores compared with the untreated ones (Fig. 3). Thymol at 250 μg/mL and carvacrol at 300 μg/mL can completely inhibit spore germination. However, the low concentration of OVEO had no significant effect on the spore germination rate of B. cinerea. The spore germination was 80.03% at the concentration of 300 μg/mL.

Figure 3 Effects of OVEO, carvacrol and thymol on inhibition of spore germination of B. cinerea.

(Columns for each concentration from left to right denote carvacrol, thymol and OVEO). The fungal germination rate of the control and treatments with different concentrations of the reagents were measured after 8 h of cultivation. The data were expressed as mean ± SD of three replicates. Values of different letters in the same column were significantly different at P < 0.05. The X-axis refers to different concentrations, and the Y-axis refers to the inhibition rate of spore germination.

Effects of carvacrol and thymol on the mycelial morphology of B. cinerea

The mycelial morphology of B. cinerea treated with carvacrol and thymol at the concentration of 40 μg/mL is observed by SEM (Fig. 4). There were regular, uniform and complete mycelia with smooth surfaces in the control group, while the mycelia treated with carvacrol and thymol showed great morphological changes, including irregular growth of mycelium, formation of verrucous surface, shrinkage, collapse and hollowing of hyphae.

Figure 4 Effects of carvacrol and thymol on the mycelial morphology of Botrytis cinerea.

Images obtained by scanning electron microscopy (Model SU8010-3400N; HITACH) with 2.5 k× magnifications at 10 kV. (A) Healthy hyphae control. (B) Hyphae treated with carvacrol at 40 μg/mL. (C) Hyphae treated with thymol at 40 μg/mL.

Effects of carvacrol and thymol on fresh and dry mycelium weight of B. cinereal

The mycelial growth of B. cinerea in PDB medium containing carvacrol and thymol for 3 d is significantly inhibited (Fig. 5). The fresh and dry weights of the mycelia treated with carvacrol were 21.93 and 5.83 mg, 8.30 and 4.13 mg, respectively and had EC50 of 9.09 μg/mL and EC90 of 40.62 μg/mL. The inhibition rates of fresh and dry weights were 86.88%, 96.51% and 78.53%, 89.31%, respectively. After being treated with thymol, the mycelial inhibitions of fresh and dry weights were 78.06% and 98.23%, 69.94% and 95.97%, respectively where the EC50 reached 21.32 μg/mL and EC90 reached 65.13 μg/mL.

Figure 5 Effects of carvacrol and thymol against mycelial biomass of B. cinerea.

Columns for each concentration from left to right denote (A) carvacrol fresh weight A, carvacrol dry weight B, thymol fresh weight C and thymol dry weight D. (B) inhibition rate of carvacrol fresh weigh A, inhibition rate of carvacrol dry weight B, inhibition rate of thymol fresh weight C and inhibition rate of thymol dry weight D. The fungal mycelial fresh and dry of the control and treatments with different concentrations reagents were measured after 3 d of shake cultivation. The data were expressed as mean ± SD of three replicates. Significant difference (P < 0.05) between the mean values was indicated by the letters above the histogram bars.

Effects of carvacrol and thymol on cell leakage of B. cinerea

A linear relative conductivity response with the increasing concentration of three reagents are found in all treatments (Fig. 6). Compared with the blank control, the two treatments with carvacrol and thymol maintained a higher level of relative conductivity. When carvacrol and thymol were at the lowest concentrations of 10 μg/mL and 70 μg/mL, at the treatment time points of 10, 40, 140, 220, 360 and 400 min, the relative conductivity values were 2.98%, 10.26%, 32.48%, 37.97%, 47.68%, 51.51% and 10.87%, 15.41%, 23.58%, 28.81%, 37.87%, 39.83%, respectively. The relative conductivity values were comparable to that of the positive control (vinclozolin) group at the same time. There was a higher relative conductivity values when carvacrol and thymol increased to 200 μg/mL and 500 μg/mL, compared with that of the vinclozolin. This indicates that there was more leakage of cell contents.

Figure 6 Effect of different concentration of carvacrol (A) and thymol (B) on membrane permeability of B. cinerea.

The fungal mycelial of the control and treatments with different concentrations reagents were measured after 108 h of shake cultivation. Values are the mean ± SD of three replicates. The X-axis refers to time, and the Y-axis refers to the relative extracellular conductivity of B. cinerea.

Effects of carvacrol and thymol on reactive oxygen species (ROS) accumulation, fungal membrane integrity and mitochondrial injury

The results of ROS accumulation, fungal membrane integrity and mitochondria damage experiments are presented in Figs. 7–9. Compared with the untreated hyphae, the treated hyphae gave off strong green fluorescence, indicating there was a large amount of ROS accumulation in the mycelium (Fig. 7). SYTOX Green penetrated the mycelium treated with carvacrol and thymol, indicating that the integrity of the mycelial membrane was damaged (Fig. 8). Compared with normal mitochondria, rhodamine 123, which entered the mitochondria from the outside, was re-released after the membrane was damaged, emitting yellow-green fluorescence (Fig. 9).

Figure 7 Effects of carvacrol and thymol on ROS measured by fluorescence microscope.

(Nikon Eclipse Ti-S, Japan) with 600× magnifications. First row: bright field. Second row: H2DCFDA. (A) and (D) untreated; (B) and (E) treated with carvacrol at 100 μg/mL; (C) and (F) treated with thymol at 100 μg/mL.

Figure 8 Effects of carvacrol and thymol on plasma membrane integrity measured by fluorescence microscope.

(Nikon Eclipse Ti-S, Japan) with 600× magnifications. First row: bright field. Second row: SYTOX green. (A) and (D) untreated; (B) and (E) treated with carvacrol at 200 μg/mL; (C) and (F) treated with thymol at 200 μg/mL.

Figure 9 Effects of carvacrol and thymol on mitochondrial integrity measured by fluorescence microscope.

(Nikon Eclipse Ti-S, Japan) with 200× magnifications. First row: bright field. Second row: rhodamine 123. (A) and (D) untreated; (B) and (E) treated with carvacrol at 200 μg/mL; (C) and (F) treated with thymol at 200 μg/mL.

Application of fungicides and disease assessment in vivo

The results of in vivo conditions showed that OVEO, carvacrol and thymol have different protective and therapeutic effects on tomato gray mold caused by B. cinerea (Table 4). At the concentration of 500 μg/mL, the protective and therapeutic effects of OVEO were significantly lower than pyrimethanil at 400 μg/mL. Although the protective effect of carvacrol was comparable to pyrimethanil, its therapeutic effect was lower. The therapeutic and protective effects of thymol were lower than that of the chemical control. At the concentration of 1,000 μg/mL, OVEO and thymol had the same protective and therapeutic effects as 400 μg/mL pyrimethanil. The protective effect of carvacrol was significantly higher than pyrimethanil while the therapeutic effect was similar to that of the chemical control. Among all the treatment groups, the protective effect of carvacrol (77.98%) was the best.

Table 4 Protective and therapeutic effects of OVEO, carvacrol and thymol against tomato gray mold cuased by B. cinerea.

Reagents	Concentration (μg/mL)	Protective effects		Therapeutic effects	
Patch diameter (cm)	Relative efficacy (%)		Patch diameter (cm)	Relative efficacy (%)	
OVEO	500	2.01 ± 0.20 b	18.88 d		2.49 ± 0.22 b	9.41 c	
1,000	1.52 ± 0.23 cd	38.50 b		2.21 ± 0.24 cd	19.70 ab	
Carvacrol	500	1.50 ± 0.23 cd	39.40 b		2.39 ± 0.28 bc	13.32 bc	
1,000	0.54 ± 0.21 e	77.98 a		1.98 ± 0.44 d	28.04 a	
Thymol	500	1.75 ± 0.23 c	29.29 c		2.47 ± 0.28 bc	10.36 bc	
1,000	1.34 ± 0.27 d	45.92 b		2.02 ± 0.19 d	26.50 a	
Pyrimethanil	400	1.41 ± 0.38 d	43.15 b		2.02 ± 0.26 d	26.56 a	
Control	0	2.47 ± 0.22 a	0.00 e		2.75 ± 0.21 a	0.00 e	
Note:

Values are expressed as the means ± SE of nine replicates. Different letters in each column indicate statistically significant differences (P < 0.05).

Discussion

It has become a trend in recent years to find biosafe antimicrobial to reduce environmental pollution and to cope with the resistance of traditional chemical agents. Many EOs and their components exhibit antifungal properties, but their high production cost and low concentration of active ingredients limit their direct use in the control of plant and animal fungal diseases. Nevertheless, the use of plant EOs to control agricultural fungal diseases has been a hot research topic, such as Cymbopogon nardus and Syringa oblata EOs as a potential biocontrol agents to control black rot in cherry tomato (Chen et al., 2014) and brown spot in tobacco caused by Alternaria alternata (Jing et al., 2018), with great application potential in controlling important pathogen diseases of crops. Moreover, it has been found that 500 μL/L mint oil and its volatile vapor at 25 °C could significantly inhibit the germination of conidia and the occurrence of disease in vivo (Xueuan et al., 2017). Eucalyptus staigeriana oil had the highest inhibitory activity against B. cinerea spores and hyphae at 0.5 μL/mL (Pedrotti et al., 2019), and α-pinene and β-caryophyllene from Cupressus sempervirens EO had high antifungal activity on B. cinerea when used it in combination or alone (Rguez et al., 2018). In addition, previous studies have found that EOs as C. cassia (Wang et al., 2014), F. vulgare (Lopez-Reyes et al., 2013), I. verum (Lee et al., 2007), M. spicata (Benomari et al., 2018), O. vulgare (Abbaszadeh et al., 2014) and S. aromaticum (Gago et al., 2019) were all showed antifungal activities in different degrees against B. cinerea. To our knowledge, in this study, the mycelial growth inhibition of L. cubeba and A. sieboldii EOs on B. cinerea have not been reported.

Phenols with free—OH group can modify various amino acid residues of proteins, and interact with potential protein targets in fungal cells (Wink, 2015). At present, more than 150 compounds, such as phenols, terpenes and their derivatives, have been reported from O. vulgare in different regions, and most of them contain different contents of carvacrol and thymol (Hernandez-Hernandez et al., 2014; De las M Oliva et al., 2015; Khan et al., 2019). In this study, GC/MS analysis showed that the main components of OVEO were carvacrol (89.98%), β-caryophyllene (3.34%) and thymol (2.39%). Previous studies have shown that carvacrol and thymol had strong inhibitory activity on a variety of pathogenic fungi and bacteria (Abbaszadeh et al., 2014; Elshafie et al., 2015; Taleb et al., 2018; Wang et al., 2019). In our experiment, carvacrol and thymol had a higher antifungal activity than OVEO on the mycelial growth, spore germination and the therapeutic and protective effects on tomato gray mold caused by B. cinerea in vitro, while their inhibitory effect was a concentration-dependent manner. Furthermore, we found that β-caryophyllene had no significant fungicidal activity on B. cinerea. This result was also confirmed by other antibacterial activity test (Purkait et al., 2020). These results demonstrated that the carvacrol and thymol, as phenol monomers, were the main fungicidal components of OVEO and the single component had a higher antifungal activity than the mixture. But whether there are synergistic effects between them and other components needs further study.

Protoplasmic membrane system, as a protective screen of cells, is necessary for the survival of fungus (Zhou et al., 2018). In this experiment, the destruction of cell membrane was confirmed by SEM and SYTOX Green fluorescence staining. Consistent with previous expectations and recent reports (Zhang et al., 2019), the results of SEM showed that the normal morphology of B. cinerea mycelia were damaged. After carvacrol and thymol treatment, the hypha appeared shrinkage, collapse and disorganization compared with that in the control. In subsequent experiments, the fluorescent microscopy observation also confirmed that carvacrol and thymol destroyed mycelial membranes of B. cinerea in this study.

Previous studies have shown that EO could lead to the rupture and damage of fungal cell membrane (Yu et al., 2015) and abnormal transport function, nucleic acid and intracellular protein leakage, and abnormal metabolism, which is the possible mechanism of antifungal action of EOs (Chen et al., 2014; Jing et al., 2018). The relative leakage test showed that 200 μg/mL carvacrol and thymol could significantly increase the relative permeability of cells and 10 mg/L carvacrol and 70 mg/L thymol at the treatment time points of (0–400 min) were equivalent to that of vinclozolin. It was speculated that carvacrol and thymol as well as vinclozolin with known mechanism of action can destroy the function of cell membrane (Choi, Lee & Cho, 1996; Cabral & Cabral, 1997) and cause leakage of enterocyte.

Reactive oxygen species (highly potent oxidants containing oxygen) encompasses oxygen free radicals and nonradical oxidants, can be interconverted from one to another by enzymatic and nonenzymatic mechanisms. A large amount of ROS can lead to oxidative damage of DNA base in cells and affect various redox reactions intra- and extracellular processes (Zorov, Juhaszova & Sollott, 2014), and play an important role in cell apoptosis (Wu et al., 2018). By using fluorescent dye DCFH-DA and flow cytometer, there was increased ROS intensity in Aspergillus flavus after the treatment of Anethum graveolens L. EO at different doses (Tian et al., 2012). Tea tree oil could lead to decreased activity of enzymes related to the tricarboxylic acid cycle and mitochondrial dysfunction as well as sharply enhanced levels of ROS in B. cinerea (Li et al., 2017). In our experiment, the H2DCFDA staining experiment showed that carvacrol and thymol significantly increased the content of ROS in mycelia. This may indirectly lead to the death of fungal cells.

Mitochondria are important organelles that produce energy and biochemical reaction sites in cells. Mitochondrial damage results in disruption of the respiratory chain and the tricarboxylic acid (TCA) cycle pathway (Fernie, Carrari & Sweetlove, 2004), induces ROS accumulation and decreases intracellular productivity (Zorov, Juhaszova & Sollott, 2014). Hu et al. (2017) found that Curcuma longa EO affected the mitochondrial ATPase and dehydrogenases activity (malate dehydrogenase, succinate dehydrogenase) in A. flavus. Through RNA-seq, it was found that the expression levels of most genes in Fusarium oxysporum after treatment with thymol, including glycolipid biosynthesis and glycolipid metabolism, were down-regulated while genes involved in antioxidant activity, chitin biosynthesis, and cell wall modification were up-regulated (Zhang, Ge & Yu, 2018). In Rhodamine-123 staining experiment, the hyphae emitted strong green fluorescence after treatment with 200 µg/mL carvacrol and thymol. It was confirmed that mitochondria were damaged. There is no doubt that mitochondrial damage accelerates the rate of fungal cell apoptosis and leads to further accumulation of ROS. However, it was not clear whether the accumulation of ROS was caused by carvacrol and thymol or indirectly by the damage of mitochondria.

Conclusion

In conclusion, this study proved that a variety of plant EOs have fungicidal potential against B. cinerea, and found that carvacrol and thymol, as the main components of OVEO, can destroy the mycelial morphology, increase cell membrane permeability, cause mitochondrial damaged and ROS accumulation in B. cinerea. It was proved that carvacrol and thymol have high potential in the control of gray mold caused by B. cinerea.

Supplemental Information

Supplemental Information 1 Inhibitory activity of 17 plant essential oils on the mycelial growth of B. cinerea.

Click here for additional data file.

Supplemental Information 2 Effects of OVEO, carvacrol and thymol on the spore germination of B. cinerea.

Click here for additional data file.

Supplemental Information 3 Effects of carvacrol and thymol against mycelial biomass of B. cinerea..

Click here for additional data file.

Supplemental Information 4 Effects of different concentrations of carvacrol, thymol, and β-caryophyllene on mycelial growth of Botrytis cinerea.

Click here for additional data file.

Supplemental Information 5 Protective and therapeutic efficacy of O. vulgare essential oil, carvacrol and thymol against tomato gray mold caused by B. cinerea.

Click here for additional data file.

Supplemental Information 6 Effect of different concentration of cavacrol (a) and thymol (b) on membrane permeability of B. cinerea.

Click here for additional data file.

Supplemental Information 7 Effects of cavacrol (a) and thymol (b) against mycelial biomass of B. cinerea (3 d).

Click here for additional data file.

Supplemental Information 8 Effects of O. vulgare essential oil, carvacrol and thymol on the spore germination of B. cinerea.

Click here for additional data file.

Supplemental Information 9 Effects of carvacrol and thymol on plasma membrane integrity measured by fluorescence microscope (Nikon Eclipse Ti-S, Japan) with 600× magnifications.

Click here for additional data file.

Supplemental Information 10 Toxicities of O. vulgare essential oil, carvacrol and thymol against mycelial growth of B. cinerea.

Click here for additional data file.

Supplemental Information 11 Effects of carvacrol and thymol on the mycelial morphology of Botrytis cinerea.

Click here for additional data file.

Supplemental Information 12 Protective and therapeutic effects of O. vulgare essential oil, carvacrol and thymol against tomato gray mold cuased by B. cinerea.

Click here for additional data file.

Supplemental Information 13 Effects of carvacrol and thymol on ROS measured by fluorescence microscope (Nikon Eclipse Ti-S, Japan) with 600× magnifications.

Click here for additional data file.

Supplemental Information 14 Effects of carvacrol and thymol on mitochondrial integrity measured by fluorescence microscope (Nikon Eclipse Ti-S, Japan) with 200× magnifications.

Click here for additional data file.

Supplemental Information 15 Inhibitory activity of 17 plant essential oils on the mycelial growth of B. cinerea.

Click here for additional data file.

Supplemental Information 16 Chemical components of the O. vulgare essential oil determined by GC–MS.

Click here for additional data file.

Additional Information and Declarations

Competing Interests

Author Contributions

Data Availability

The authors declare that they have no competing interests.

Huiyu Hou conceived and designed the experiments, performed the experiments, analyzed the data, prepared figures and/or tables, authored or reviewed drafts of the paper, and approved the final draft.

Xueying Zhang performed the experiments, prepared figures and/or tables, and approved the final draft.

Te Zhao analyzed the data, authored or reviewed drafts of the paper, and approved the final draft.

Lin Zhou conceived and designed the experiments, analyzed the data, authored or reviewed drafts of the paper, and approved the final draft.

The following information was supplied regarding data availability:

The raw measurements are available in the Supplemental Files.

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
