# Peer review of "Effects of Origanum vulgare essential oil and its two main components, carvacrol and thymol, on the plant pathogen Botrytis cinerea"

_PeerJ, doi:10.7717/peerj.9626_

## Round 0.1 · original submission · Major Revisions

Dear authors
check the comments of the reviewers.
As I worked on essential oils and their antimicrobial activity, I would add more recommendations:

1. GC-MS. Show GC Profile and label the peaks
2. Include MS data in the table
3. antifungal activity; you need a positive control (e.g. a know antifungal compound); without it, the results are useless
4. thymol and carvacrol have been studied intensively; they are phenolic monoterpenes and therefore more active than others
read
Wink, M. 2015. Modes of Action of Herbal Medicines and Plant Secondary Metabolites. Medicines, 2, 251-286.
5. your essential oils have a moderate but not a strong activity
6. Literature data with IC50 below 1 µg/ml are wrong; the authors added microliters in the assay. when they converted it to µg they wrongly assumed that 1 µl = 1 µg, but it is 1 mg
7. your literature survey on Origanum is superficial. This plant has been studied in Europe for many years and many papers have been published; papers on antibacterial activity usually also have data on fungi - check
Regards
M. Wink

AE

Reviewer 1 ·

Basic reporting

ok

Experimental design

need some corrections as revised.

Validity of the findings

need some modifictaion in the discussion part

Additional comments

The revision has been made as track revision in the attached word file.

Annotated reviews are not available for download in order to protect the identity of reviewers who chose to remain anonymous.

Reviewer 2 ·

Basic reporting

Please see all my comments under section 3

Experimental design

Please see all my comments under section 3

Validity of the findings

Manuscript # 45399
Title: Effects of Origanum vulgare essential oil and its two main components, carvacrol and thymol, on the plant pathogen Botrytis cinerea
Authors: Huiyu Hou, Lin Zhou, Xueying Zhang, Te Zhao

In this study, the authors examined the impact of Origanum vulgare essential oil and its components on the plant pathogen Botrytis cinerea. The study, overall, is well designed and has strong potential in expanding our understanding of how these essential oil components may be used to control the pathogen and its spread. The manuscript is well written - clear, professional and articulate English has been used throughout. Authors have used good literature citations, and sufficient background and rationale for the work has been included in the manuscript.

I have the following comments about the manuscript, please see below.

Introduction, Lines 55-56 – Please elaborate on “and the point mutations have been identified”.

Introduction, paragraph 3, Line 72 – Please elaborate of what “bioactivities”. Please also cite some articles on this.

Materials and methods, Section 1, Inhibitory activity of different plant essential oils on mycelium growth of B. cinerea – Please spell out PDA (I am assuming this is Potato Dextrose Agar) on first use. On paragraph 2 of this section, can you please put a citation here for this method?
Materials and methods, Section 3, In vitro antifungal activities of O. vulgare essential oil, carvacrol and thymol against B. cinerea – With ‘mother liquor”, do you mean a combined solution? Please change this term with a more scientific usage.

Materials and methods, Section 4, Antifugal activity of O. vulgare essential oil, carvacrol and thymol on spore germination – “The spore concentration was adjusted to..” – how did you adjust spore concentration? Please elaborate and include a citation to the method here.

Materials and methods, Section 5, Effects of carvacrol and thymol on fresh and dry mycelial weight of B. cinerea – Please spell our PDB on first use (I am assuming you mean Potato Dextrose Broth?)

Results, Inhibitory activities of 17 plant essential oils on mycelial growth of B. cinerea – I suggest putting the percent inhibition rates on a plot for better understanding. The table is useful, if needed it could be put into a supplement.

Results, GC-MS analysis of O. vulgare essential oil – Is this table 2? Do you have the GC-MS peaks images/plots from this analysis? I suggest using a figure with the peaks rather than a table here.
Results, Antifungal activities of the O. vulgare essential oil, carvacrol and thymol against mycelial growth of B cinerea in vitro – I suggest the same here, please include a figure here instead of a table.

Results, Antifugal activities of O. vulgare essential oil, carvacrol and thymol on spore germination - I suggest the same here, this data is better depicted in a plot to include as a figure here instead of a table.

Results, Antifungal activities of O. vulgare essential oil, carvacrol and thymol on spore germination of B. cinerea - I suggest the same here, this data is better depicted in a plot to include as a figure here instead of a table.

Results, Effects of carvacrol and thymol on the mycelial morphology of B. cinerea - I suggest the same here, this data is better depicted in a plot to include as a figure here instead of a table.

Results, Effects of carvacrol and thymol on wet and dry mycelium weight of B. cinerea – I suggest you also include a plot with inhibition rates here (this could potentially be a supplemental figure but including a plot lays out your findings well)

Figures 2 and 3: On the legend, please explain your X and Y axes.

There is not Figure 7 cited within the submitted manuscript, although there is one included at the end of it.

Discussions, Line “a hot-topic with great application potential in controlling important fungal diseases of crops – this needs more elaboration with a few citations.

·

Basic reporting

I have recommended some language corrections in the attached file
I have some comments in the from of queries for the authors which I wish they answer
There are two refrences missing form the list of refrences

Experimental design

no comments

Validity of the findings

no comments

Additional comments

All corrections and comments are embded to the attached file

---

## Round 0.2 · accepted · Accept

Dear authors,

Your revision is adequate. Congratulations- your manuscript is accepted

Regards
Michael Wink
AE

Reviewer 2 ·

Basic reporting

Authors made necessary changes to the manuscript based on my previous review suggestions.

Experimental design

no comment

Validity of the findings

no comment